# A Physiological Approach to Explore How Thioredoxin–Glutathione Reductase (TGR) and Peroxiredoxin (Prx) Eliminate H_2_O_2_ in Cysticerci of *Taenia*

**DOI:** 10.3390/antiox13040444

**Published:** 2024-04-10

**Authors:** Alberto Guevara-Flores, Gabriela Nava-Balderas, José de Jesús Martínez-González, César Vásquez-Lima, Juan Luis Rendón, Irene Patricia del Arenal Mena

**Affiliations:** 1Departamento de Bioquímica, Facultad de Medicina, Universidad Nacional Autónoma de México (UNAM), Apartado Postal 70-159, Mexico City 04510, Mexico; guevarafa@bq.unam.mx (A.G.-F.); jjmtz@bq.unam.mx (J.d.J.M.-G.); cesarlima@ciencias.unam.mx (C.V.-L.); jrendon@bq.unam.mx (J.L.R.); 2Departamento de Microbiología y Parasitología, Facultad de Medicina, Universidad Nacional Autónoma de México (UNAM), Apartado Postal 70-159, Mexico City 04510, Mexico

**Keywords:** peroxiredoxins, *Taenia crassiceps*, thioredoxin–glutathione reductase, hydrogen peroxide

## Abstract

Peroxiredoxins (Prxs) and glutathione peroxidases (GPxs) are the main enzymes of the thiol-dependent antioxidant systems responsible for reducing the H_2_O_2_ produced via aerobic metabolism or parasitic organisms by the host organism. These antioxidant systems maintain a proper redox state in cells. The cysticerci of *Taenia crassiceps* tolerate millimolar concentrations of this oxidant. To understand the role played by Prxs in this cestode, two genes for Prxs, identified in the genome of *Taenia solium* (*Ts*Prx1 and *Ts*Prx3), were cloned. The sequence of the proteins suggests that both isoforms belong to the class of typical Prxs 2-Cys. In addition, *Ts*Prx3 harbors a mitochondrial localization signal peptide and two motifs (-GGLG- and -YP-) associated with overoxidation. Our kinetic characterization assigns them as thioredoxin peroxidases (TPxs). While *Ts*Prx1 and *Ts*Prx3 exhibit the same catalytic efficiency, thioredoxin–glutathione reductase from *T. crassiceps* (*Tc*TGR) was five and eight times higher. Additionally, the latter demonstrated a lower affinity (>30-fold) for H_2_O_2_ in comparison with *Ts*Prx1 and *Ts*Prx3. The *Tc*TGR contains a Sec residue in its C-terminal, which confers additional peroxidase activity. The aforementioned aspect implies that *Ts*Prx1 and *Ts*Prx3 are catalytically active at low H_2_O_2_ concentrations, and the *Tc*TGR acts at high H_2_O_2_ concentrations. These results may explain why the *T. crassiceps* cysticerci can tolerate high H_2_O_2_ concentrations.

## 1. Introduction

Reactive oxygen species (ROS) including the superoxide anion, the hydroxyl radical, the O_2_ singlet, and hydrogen peroxide (H_2_O_2_) are among the compounds resulting from aerobic metabolism. H_2_O_2_ possesses characteristics that reveal its relevance inside cells, such as the following: (I) it has no charge; (II) it is a very stable molecule compared to other ROS, and consequently, it has the longest half-life; (III) it has the highest diffusion rate, which allows it to diffuse in the whole cell; and (IV) at low concentrations, it acts as a second messenger in signaling pathways [1,2]. To avoid the deleterious accumulation of H_2_O_2_, organisms rely upon diverse metal-dependent peroxidases, including catalase (CAT) [3] and two thiol-dependent (-SH) antioxidant systems: (a) the glutathione system, composed of glutathione tripeptide (GSH), glutathione reductase (GR), and glutathione peroxidase (GPx), and (b) the thioredoxin system, composed of the small protein thioredoxin (Trx), thioredoxin reductase (TrxR), and peroxiredoxin (Prx). Both thiol-dependent antioxidant systems require NADPH [4]. Together, these systems regulate the H_2_O_2_ concentration, which maintains an adequate intracellular redox homeostasis in most organisms [5]. It is important to note that CAT is usually confined to peroxisomes [6], and in many endoparasitic organisms like cestodes, this enzyme is absent [7]. In contrast, GPxs and Prxs are present in most organisms, with different isoforms found in diverse cell compartments [8]. Prxs are characterized mainly by two points: (1) their catalytic efficiency (*k*_cat_/*K*_m_) for H_2_O_2_ is lower (10^4−5^ M^−1^ s^−1^) [9,10,11], and this low catalytic efficiency is compensated by (2) their high intracellular concentration that ranges between 15 and 60 μM [12].

All Prxs depend on the presence of a catalytic cysteine around position 50 (Cys^~50^) that reacts with H_2_O_2_; identified as peroxidatic cysteine (C_P_SH). Based on this, Prxs are most often classified by the number of catalytic cysteine residues per subunit. Prxs with one cysteine (Prx 1-Cys) and two cysteines (Prx 2-Cys) exist. For Prx 2-Cys, the second cysteine (Cys^~170^) was identified as the resolving cysteine (C_R_SH) [13,14]. The reduction of H_2_O_2_ is performed through the oxidation of C_P_SH to sulfenic acid (CpSOH); subsequently, this sulfenic acid reacts with C_R_SH, generating a disulfide bond (C_P_S-SC_R_). When this disulfide bond is intermolecular, the Prxs are “typical”, and when it occurs in the same subunit, they are described as “atypical” [15]. In both cases, the disulfide bond is generated anew to its dithiol form by the reduced forms of thioredoxin (Trx-(SH)_2_) or glutathione (GSH) [16]. A shared feature by most Prxs is that they are sensitive to overoxidation, with micromolar concentrations of H_2_O_2_, and are known as “sensitive Prxs” [16,17,18]. In sensitive Prxs, two structural motifs (-GGLG- and -YP-) have been described that are predicted to confer sensitivity to H_2_O_2_. These sites are highly conserved among the Prxs of eukaryote cells; however, recently, “robust Prxs” (resistant to overoxidation) have been reported in bacteria including *Escherichia coli* and *Salmonella*, which lack these motifs and instead harbor two highly conserved motifs that have been associated with resistance to H_2_O_2_ [19]: (A and B: -D(N/G)H(G/S)- and -T(S/T)-, respectively).

The enzymatic activity of Prxs was determined with an assay coupled with TrxR and Trx using as reducer to NADPH. Generally, for this assay, the enzyme coupling of *E. coli* [20,21] and yeast [22] are the most used. These organisms’ reductases lack a Sec residue, so their TrxR is termed TrxR-Cys. Markedly, endogenous proteins were used to determine the activity of Prxs for a few organisms, like *Plasmodium falciparum* (*Pf*TrxR-Cys and *Pf*Trx) [23]. The eukaryotic TrxRs are selenocysteine-dependent (termed TrxR-Sec) and generally have the capacity to recognize Trxs of another origin as substrates [24], whereas the TrxR-Cys of prokaryotes are usually highly specific for their own endogenous Trx [24,25]. On the other hand, the specificity of Prxs for Trxs of other origins is not well documented. This information is relevant to establish which system is more appropriate to determine the activity of Prxs in a physiological context.

Parasite plathelminths of the cestode class must have a robust mechanism for the depuration of ROS that are either generated by the host’s immunological system [26] or from their own metabolism [26,27]. Studies performed in *Taenia crassiceps* have demonstrated significant amounts of H_2_O_2_ production under basal conditions [28], and the larval form can tolerate exposure to higher concentrations of H_2_O_2_ in culture conditions [29]. However, it is widely documented in diverse parasitic platyhelminths at both the genomic and proteomic levels that cestodes lack CAT, TrxR, and GR [7,30]; hence, their redox homeostasis relies on a bifunctional enzyme: the thioredoxin–glutathione reductase (TGR-Sec), which is the sole enzyme responsible for maintaining both thioredoxin and glutathione in their reduced state. Regarding the thiol-dependent peroxidases, a gene that encodes a GPx has been previously described in the *Taenia solium* genome, which is predicted to be associated with the plasma membrane [31], as well as two genes that encode 2-Cys Prxs isoforms [31].

This work aimed to identify the factors involved in the high tolerance of the *Taenia* genus to millimolar concentrations of H_2_O_2._ In this study, two Prxs of the *T. solium* cysticerci were cloned and expressed. Here, we characterize how they remove H_2_O_2_ using their endogenous thioredoxin system and the role of *T. crassiceps* cysticerci (*Tc*) TGR-Sec in this process.

## 2. Materials and Methods

### 2.1. Chemicals

NADPH, H_2_O_2_, ter-butyl hydroperoxide solution (Luperox), cumene hydroperoxide, Trizol^®^, bacto yeast, bacto tryptone, IPTG, ampicillin, and chloramphenicol, as well as Tris, EDTA, oxidized glutathione (GSSG), reduced glutathione (GSH), PMSF, manganese (II) chloride, L-glutamine, hydroxylamine, ADP, DEAE-cellulose, HA-Ultrogel, and Cibacron Blue 3G-A were obtained from Sigma-Aldrich, Merck KGaA, (Darmstadt, Germany). All other chemicals were purchased from JT Baker Chemical, Phillipsburg, NJ, USA.

### 2.2. Biological Material

*T. solium* cysticerci were obtained from the skeletal muscle of naturally infected pigs from City of Cuautla, State of Morelos, México. The cysticerci were washed with phosphate buffer (PBS), pH 7.4, and frozen until use. *T. crassiceps* cysticerci (HYG strain) were obtained from the peritoneum of experimentally infected BALB/c mice as described [32], washed with PBS, and frozen until use. All animal care and research protocols were carried out in accordance with the guidelines for the ethical care of experimental animals according to the guidelines of the Official Mexican Standards for the production, care, and use of laboratory animals (NOM-062-ZOO-1999). Further, the experimental protocols reported in the present work were approved by the Internal Committee for the Care and Use of Laboratory Animals (CICUAL) of the Facultad de Medicina, Universidad Nacional Autónoma de México (008-CIC-2023). All efforts were made to minimize animal suffering and to reduce the number of animals used.

### 2.3. Cloning and Overexpression of TsPrx1 and TsPrx3

Plasmid pET-23a (+) was obtained from Novagen^®^, Merck KGaA group (Darmstadt, Germany). *E. coli* strains TOP10 and BL-21 Codon Plus (DE3) were purchased from the Invitrogen corporation (Carlsbad, CA, USA). The plasmid purification kit was obtained from Thermo-Scientific (Waltham, MA, USA), as were the NdeI and Xho I restriction enzymes and the RevertAid First Strand synthesis kit used to obtain the cDNA. The amplified (TAQ DNA polymerase) was obtained from BioTecMol (Mexico City, Mexico). T4 DNA ligase was purchased from Promega Corporation (Madison, WI, USA), and the GelRed^®^ was obtained from Biotium (Fremont, CA, USA). Trx from *E. coli*, Trx from humans, TrxR from rats, TrxR from *E. coli*, and GR from *yeast*, were obtained from Sigma-Aldrich.

Two Prxs coding sequences were identified in the WormBase Parasite (https://parasite.wormbase.org/Taenia_solium_prjna170813/Info/Index/, accessed on 2 August 2022): *Ts*Prx1 (22 kDa) [33] and *Ts*Prx3 (25 kDa). The total RNA from three *T. solium* cysticerci was extracted with TriZol^®^, and the cDNA synthesis was carried out using the RevertAid First Strand synthesis kit (Thermo Scientific, Waltham, MA, USA) with the supplier’s specifications and using oligo (dT)_12_ primer (5 μM final concentration). The synthesized cDNA (2 μL) was used as a template to amplify the Prxs genes by means of PCR reactions (50 μL total volume), using 100 ng/μL (0.2 μM) of each oligonucleotide 5′-ATTCATATGGCTGCTGCTGTCATCGGG-3′ and 3′-AAACTCGAGTCTTGAGCTCATGAACGAC-5′ for the TsPrx1 isoform; for *Ts*Prx3, the oligonucleotides 5′AAGCATATGCAGCGTCTTATGCCTCATC-3′ and 3′TATCTCGAGGTTGACCTTCTCAAAGTACGC-5′ were used. The PCR reactions were carried out at an initial incubation temperature of 94 °C for 30 s; the alignment temperature was 61 °C for 35 s, and the extension temperature was 72 °C for 90 s; the final extension temperature was 72 °C for 10 min. PCR products were analyzed by electrophoresis on a 1.5% agarose gel with known molecular weight (MW) markers and visualized with GelRed^®^ at λ = 312 nm; the products were purified and sequenced via the Sanger method [34] at the Sequencing Unit of the Institute for Biomedical Research (IIB, Cuernavaca, Morelos, Mexico). The resultant sequences were aligned and compared with the sequences of *Ts*Prx1 and *Ts*Prx3 identified in the GeneBank database, using the NCBI BLAST page (https://blast.ncbi.nlm.nih.gov), accessed on 29 September 2023. 

The amplified *Ts*Prx1 and *Ts*Prx3 genes were cloned into the pET-23a(+) expression vector (Novegen, Dublin, Ireland), using the NdeI and XhoI cutting sites. The constructs were used to transform *E. coli* TOP 10 and Codon Plus bacteria. Positive clones were identified by PCR reactions with the specific oligos. Codon Plus positive bacterial clones were grown in LB culture (Luria–Bertani) with ampicillin (0.1 mg/mL) and chloramphenicol (34 μg/mL). The induction of the expression of clones *Ts*Prx1 and *Ts*Prx3, both with His tags in their amino terminal ends, was carried out by adding 1 mM IPTG at 37 °C and 300 rpm. After 4 h of incubation, bacteria were recovered by centrifugation and lysed by sonication at a frequency of 20 KHz. The expression of *Ts*Prx was confirmed by SDS-PAGE according to Laemmli [35] and stained with Coomassie blue. The Prxs were purified from the soluble bacterial lysate by affinity chromatography on IMAC Sepharose (BioRad, Hercules, CA, USA). The protein concentration was determined with the extinction coefficient (ε) of each protein [36].

### 2.4. Purification of the TGR from T. crassiceps and Recombinant Trx from T. solium

The protocol followed in the purification of cytosolic TGR from *T. crassiceps* has been described elsewhere [37], using 20 infected mice (around 400 cysticerci per mouse). The recombinant Trx from *T. solium* was obtained following the protocol previously described [38].

### 2.5. Bioinformatics Analysis

The amino acid sequence alignment of *Ts*Prx1 and *Ts*Prx3 was performed using the Clustal Omega program (https://www.uniprot.org/align/, accessed on 19 October 2022). The prediction of the subcellular location of an N-terminal peptide corresponding to *Ts*Prx3 was performed using DeepLoc-1.0 (https://services.healthtech.dtu.dk/services/DeepLoc-1.0/, accessed on 6 June 2023) and resulted in being mitochondrial-directed.

### 2.6. Electrophoresis

Polyacrylamide gel (4, 10, and 16%) electrophoresis under denaturing conditions was performed as described by Shägger [39]. Gels were stained by conventional procedures. The purity degree of the *Ts*Prx1, *Ts*Prx3, Trx proteins from *T. solium* and the TGR from *T. crassiceps* was established by analyzing the densitometry of each protein in the SDS-PAGE using ImageJ (https://imagej.nih.gov/ij/), accessed on 8 November 2023.

### 2.7. Protein Determination

The concentration of *Ts*Prx1, *Ts*Prx3, and *Ts*Trx was determined by measuring their absorbance at 278 nm. The corresponding extinction coefficients (ε) were as follows: *Ts*Prx1 = 20.6 mM^−1^ cm^−1^; *Ts*Prx3 = 21.4 mM^−1^ cm^−1^; and *Ts*Trx = 7.8 mM^−1^ cm^−1^. For *Tc*TGR, its protein concentration was determined at 460 nm based on its FAD content (ε = 11.3 mM^−1^ cm^−1^). The protein concentration was corroborated by the densitometric method [40].

### 2.8. Enzyme Assays

This section pertains to the thioredoxin reductase activity of *Tc*TGR. The reductase activity was determined by following the decrease in absorbance at 340 nm due to the oxidation of NADPH (150 μM) in the presence of recombinant *Ts*Trx. Assays were performed at 25 °C in 100 mM Tris-HCl buffer (pH 7.8) containing 1 mM EDTA (TE buffer) in a final volume of 0.6 mL. The reaction was started by adding insulin (to recycle *Ts*Trx) at a final concentration of 25 μM. An extinction coefficient of 6.22 mM^−1^ cm^−1^ for NADPH was used for the calculations of enzyme activity, as described previously [37].

Here. the activity of the peroxiredoxins from *T. solium* is discussed. This activity was determined by either of the two methods described below. The final volume of the reaction mixture was 0.3 mL. Unspecific rates were subtracted from the specific rates. All activity assays were carried out in a UV/Vis spectrophotometer DU-730 from Beckman Coulter.

#### 2.8.1. Peroxidase Activity Assays

The reductase activity of the recombinant Prxs using either H_2_O_2_ or organic hydroperoxides (cumene hydroperoxide and *t*-butyl hydroperoxide) as oxidizing substrates was determined in TE buffer by following the oxidation of 150 μM NADPH at 340 nm and 25 °C in a coupled assay with *Tc*TGR (11.2 nM), *Ts*Prx1 or *Ts*Prx3 (1.25 μM) and recombinant *Ts*Trx (60 μM), and the latter was tested as a reductant substrate; under these conditions, when the maximum reduction was obtained (baseline), the specific reaction was started by adding the corresponding peroxide. One unit of Prx activity was defined as the amount of enzyme required to cause the oxidation of 1 nmol of NADPH per minute under the assay conditions described. Alternatively, the peroxidase activity with GSH as the reductant was assayed with GR from *Saccharomyces cerevisiae* or *Tc*TGR (own reductase). The reaction mixture contained the following: (a) 0.1 unit/mL *Sc*GR (Sigma) or (b) 11.2 nM *Tc*TGR, 150 μM NADPH, 1.25 μM of *Ts*Prx1 or *Ts*Prx3, and 1 mM GSH in a buffer containing 100 mM sodium phosphates (pH 7.0), 1 mM EDTA. The reaction was initiated by adding the corresponding peroxide, and the consumption of NADPH was recorded by following the decrease in absorbance at 340 nm and 25 °C.

#### 2.8.2. Thioredoxin Peroxidase Activity of TcTGR and EcTrxR

The comparison of a selenocysteine-dependent enzyme (TcTGR) with a Cys-dependent enzyme (EcTrxR), regarding its ability to catalyze the Trx-dependent reduction of H_2_O_2_, was evaluated by mixing 150 μM NADPH with either 60 μM TsTrx and 11.2 nM TcTGR or 6 μM EcTrx and 83 nM EcTrxR in TE buffer. The reaction was started by adding 1 mM H_2_O_2_, and the absorbance at 340 nm was measured. The final volume of the reaction mixture was 0.6 mL.

The kinetic constants *K*_m_ and *k*_cat_ of *Ts*Prx1 and *Ts*Prx3 for either H_2_O_2_, *t*-butyl hydroperoxide, or the cumene hydroperoxide substrates were determined by varying the concentration of the corresponding peroxide at a constant concentration of both NADPH (150 μM) and *Ts*Trx (60 μM). To obtain the kinetic parameters for *Ts*Trx, a constant concentration of 50 μM H_2_O_2_ was used at varying *Ts*Trx concentrations. In all cases, fixed concentrations of *Ts*Prxs (1.25 μM) and *Tc*TGR (11.2 nM) were used (these last concentrations were previously determined to prevent them being limiting). The kinetic constants of *Tc*TGR toward H_2_O*_2_* was obtained by varying the concentration of the peroxide at a constant concentration of NADPH (150 μM) and *Ts*Trx (60 μM). All initial velocity data were fitted to the Michaelis–Menten equation through non-linear regression analysis using Sigma-Plot Software version 12.

#### 2.8.3. Glutamine Synthetase Protection Assay

The ability of cytosolic TcTGR, TsPrx1, and mitochondrial TsPrx3 to protect glutamine synthetase (GS) from oxidation was performed as previously described [41]. For both TsPrx1 and TsPrx3, the inactivation mixture contained 0.15 μM GS from E. coli, 3 μM FeCl_3_, and 10 mM DTT either in the presence or in the absence of 1.25 μM of the corresponding Prx in 50 mM HEPES buffer (pH 7). The final volume of the mixture was 50 μL. For TcTGR, the inactivation mixture additionally contained 160 μM NADPH and recombinant TsTrx either in the presence or in the absence of TcTGR. After 15 min of incubation at 30 °C, the residual activity of GS was determined by adding 1 mL of the assay mixture (0.4 mM ADP, 0.15 M glutamine, 10 mM Na_2_HAsO_4_, 20 mM NH_2_OH, and 0.4 mM MnCl_2_ in 100 mM HEPES buffer), pH 7.4. The resultant solution was incubated for 30 min at 30 °C; then, the reaction was terminated by adding 0.25 mL of stop mixture (0.3 M FeCl_3_ and 5.8 M HCl), and the formation of the γ-glutamylhydroxamine-Fe^3+^ complex was measured at 540 nm.

### 2.9. Data Presentation and Statistical Analysis

The data shown below represent the mean ± S.D. of three independent experiments. Data were evaluated for statistical significance using Student’s *t*-test and Statistical Software OriginPro (version 8, OriginLab Corporation, Northampton, MA, USA). 

## 3. Results

### 3.1. Recombinant Peroxiredoxins

Two Prxs of *T. solium* cysticerci were cloned and expressed: *Ts*Prx1 as described in [33] and *Ts*Prx3. The Prx1 gene has been reported previously in the *T. solium* genome with the number TsM_001155200. This gene is 655 bp in length with two exons, the first 134 bp, and the second one 454 bp with a 67 bp intron. The two exons encode a protein sequence of 195 residues. The sequence reported for a second peroxiredoxin (*Ts*Prx3) in the WormBase indicates a length of 2176 bp constituted by four exons of 357, 235, 30, and 86 bp and three introns with a length of 80, 1150, and 238 bp. The splicing of the two exons encodes a sequence of 235 amino acid residues. However, in the present work, a sequence of 224 amino acid residues was obtained because exon 3 is fused with introns 2 and 3 so that the gene for *Ts*Prx3 is constituted by three exons of 357, 235, and 86 bp and two introns of 80 and 1418 bp.

Figure 1 depicts the sequences. *Ts*Prx1, previously reported by Molina-López et al. [33], corresponds to a cytosolic Prx with a MW of ~22 kDa, whereas *Ts*Prx3 has a MW of ~25 kDa. These two proteins are 56.4% identical. *Ts*Prx3 has 29 additional residues in its N-terminal end. It was therefore analyzed with the DeepLoc-1.0 program to pinpoint its subcellular location.

This peptide resulted a mitochondrial recognition signal with a probability of ~0.44 (Appendix A). A similar result (0.57) was obtained with the hierarchical tree method (Appendix A). Both results suggest that it must correspond to a mitochondrial isoform (*Ts*Prx3), comparable to that reported for the mitochondrial Prxs of *Haemonchus contortus* and *Caenorhabditis elegans*, which also have a signaling peptide [42]. Both Prxs can be classified as “typical” because of the presence of two essential resolving cysteines: the peroxidatic cysteine^49^ (C_P_) localized in the N-terminal end and the cysteine^170^ (C_R_) located in the C-terminal end. A relevant difference between the two Prxs is the additional presence of two motifs (-GGLG-) and (-YF-) in the *Ts*Prx3 isoform (Figure 1), suggesting a possible higher sensitivity to H_2_O_2_.

### 3.2. Purity Degree of Recombinant Proteins

The degree of purity of *Ts*Prx1 and *Ts*Prx3 and *Ts*Trx recombinant proteins, and that of *Tc*TGR, was determined through electrophoresis in denaturing conditions (SDS-PAGE). Figure 2 shows that the four proteins had a significant degree of purity, which was confirmed through densitometry analysis of each band, revealing a purity greater than 75% for all proteins.

### 3.3. Peroxidase Activity of the Recombinant TsPrx1 and TsPrx3

Based on results (see below), 60 μM of *Ts*Trx was used in the activity assays, one and a half times the *K*_m_ for *Tc*TGR. For GSH, 1 mM of GSH was used, which corresponds to the concentration reported in *T. crassiceps* cysticerci [27]. The activity of *Ts*Prx1 and *Ts*Prx3 was determined by changing the concentration of H_2_O_2_ (Table 1).

### 3.4. Kinetic Analysis of TcTGR

Kinetic constants of *Tc*TGR were determined using *Ts*Trx as a substrate with the following results: *K*_m_ = 41.5 μM and *k*_cat_/*K*_m_ = 1.2 × 10^6^ M^−1^ s^−1^ (Appendix A); despite having different *K*_m_, the catalytic efficiency values were comparable to those reported previously [43] and those reported for *Ts*TGR and the recombinant *Ts*Trx [38]. Additionally, the comparison of the *Ts*TGR gene (ID: TsM_000506200) of the *T. solium* genome submitted in WormBase Parisite database (GENOME ID: PRJNA170813) and the *Tc*TGR gene (ID: JAKROA010000003.1) submitted in the GenBank database (GENOME ID: GCA_023375655.1.) showed an identity above 90%, and the genomic sequences of the *Ts*Prx1 and *Tc*Prx1 genes showed 94% identity [29]. These data suggest that independently of the origin of the proteins used in the activity assays, either of *T. solium* or *T. crassiceps*, the kinetic parameters were within the same range, and the high rates of identity of the sequences protein or genomics of the different components of the thioredoxin system (TS) enabled us to use *Tc*TGR and the recombinant *Ts*Trx with confidence in our assays.

Unexpectedly, in Appendix A, it is shown that when using GSH as substrate, it was not possible to detect the peroxidase activity in *Ts*Prx1 and *Ts*Prx3 in the presence of *Sc*GR or *Tc*TGR in the coupled assay (as mentioned under Materials and Methods Section 2). Table 1 and Appendix A show that the two Prxs depict a high affinity for different peroxides (*K*_m_ < 8.4 μM), except for *Ts*Prx1, whose affinity for the *t*-butyl hydroperoxide was significantly lower (*K*_m_ = 18.1 μM). The catalytic efficiency for the different peroxides was about ~10^4^ M^−1^ s^−1^; these kinetic parameters were within the same order of magnitude as other Prxs [10,11].

Peroxidase activity was not detected using GSH and with other organic peroxides as oxidizing substrates. Afterward, the kinetic constants for both Prxs toward the Trx were determined at a constant concentration of 50 μM of H_2_O_2_. The results obtained are shown in Table 2. It is interesting to point out that the affinity of *Ts*Prx3 for *Ts*Trx was significantly lower compared to that of *Ts*Prx1. Again, GSH was not efficient as a reducer.

### 3.5. Dependence of the Peroxidase Activity of TsPrx1 and TsPrx3 on the H_2_O_2_ Concentration

As mentioned, only *Ts*Prx3 has the two motifs that provide sensitivity to H_2_O_2_ in its sequence. To determine the susceptibility of both Prxs to H_2_O_2_, peroxidase activity was analyzed with a wide range of H_2_O_2_ concentrations. Figure 3 shows the saturation curves of both enzymes with a clear biphasic pattern, suggesting the presence of two components with peroxidase activity. A comparison of the two activity profiles reveals that the apparent maximal velocity of the component with the highest affinity is higher for the assays with *Ts*Prx1.

However, in both cases, the total maximal velocity is essentially identical. A non-linear regression analysis yielded the corresponding kinetic parameters for both systems (Table 3). Because *Tc*TGR is present as an auxiliary enzyme in the activity assays of both *Ts*Prxs, it is possible that one of the components observed in the saturation graphs could be due to *Tc*TGR. Consequently, the potential activity of the peroxidase of *Tc*TGR was analyzed in the absence of *Ts*Prxs (Appendix A). The results revealed that the peroxidase activity of *Tc*TGR is significant (*K*_m_: 79.8 μM), overlapping with the activity observed in the assays performed in the presence of *Ts*Prx1. Therefore, it can be concluded that the main contribution of the peroxidase activity is exerted by *Tc*TGR, particularly at high H_2_O_2_ concentrations. Despite its significantly lower affinity for the peroxide, the catalytic efficiency of *Tc*TGR is approximately five and eight times higher than that of *Ts*Prx1 and *Ts*Prx3, respectively (Table 3).

### 3.6. Peroxidase Activity of TrxR of E. coli

To determine whether the selenocysteine (Sec) residue plays a critical role in the high peroxidase activity of the TGR, its activity was compared to that of a TrxR lacking such residue, using the enzyme of *E. coli*. The results (Figure 4) revealed that the peroxidase activity of *Tc*TGR was significantly higher (9.4 μmol min^−1^ mg^−1^) compared with that of *Ec*TrxR (0.47 μmol min^−1^mg^−1^).

### 3.7. Protection of the Glutamine Synthetase

The peroxidase activity present in the *Tc*TGR or *Ts*Prxs and their consequent protective activity of the GS from ROS was compared. As shown in Figure 5A, in the presence of TS (NADPH + *Ts*Trx + *Tc*TGR), ~50% protection was obtained. The addition of *Ts*Prx1 resulted in ~80% protection, whereas *Ts*Prx3 did not protect and had the same magnitude regarding protection as TS (Figure 5B).

## 4. Discussion

Peroxiredoxins, enzymes that reduce H_2_O_2_, are widely represented among organisms [44]. A search in the *T. solium* genome revealed that this parasite possesses two peroxiredoxins: *Ts*Prx1 and *Ts*Prx3. The sequence analysis of both Prxs was performed, and the *Ts*Prx3 sequence showed the presence of a signaling peptide, suggesting its localization to mitochondria (Appendix A). Both sequences indicated that they could be classified within the “typical Prx 2-Cys” group (Figure 1). The *Ts*Prx3 isoform harbored the motifs (-GGLG-) and (-YP-) associated with the hyperoxidation produced by H_2_O_2_ [16,17]. Interestingly, the presence of these motifs in *Ts*Prx3 did not confer a higher or lower kinetic behavior compared to *Ts*Prx1. Both peroxiredoxins could recognize H_2_O_2_ with a catalytic efficiency of ~10^4^ M^−1^ s^−1^ (Table 1), which is comparable to other organic peroxides (Appendix A) used in the present work as substrates.

TsPrx’s affinity for H_2_O_2_ is clearly higher if compared with the Prxs of the *Schistosoma mansoni* trematode [20]. However, its catalytic efficiency is comparable to those reported for the Prxs of diverse organisms, such as the *H. contortus* nematode [42], *Bacillus subtilis* [45], and *Helicobacter pylori* [46], which reduce H_2_O_2_ using only Trx-(SH)_2_ and do not recognize GSH. In contrast, Prxs that can use both GSH and Trx-(SH)_2_ have been reported in *P. falciparum* [23,47], *S. mansoni* [20], and *Clonorchis sinensis* [21]. It is interesting to point out that among Prxs that use both reducing substrates, some, including *P. falciparum* and *S. mansoni*, use GSH more efficiently as a substrate. The results of this work indicate that GSH cannot serve as a reducing substrate and therefore is a marked preference for Trx-(SH)_2_; hence, we suggest that both *Ts*Prx1 and *Ts*Prx3 must be considered true thioredoxin peroxidases (TPx).

In Prx 2-Cys the C_P_SH thiol can reach different states of oxidation by reacting sequentially with one, two, or three H_2_O_2_ molecules, giving rise sequentially to sulfenic (C_P_SOH), sulfinic (C_P_SO_2_H), and sulfonic (C_P_SO_3_H) acids. The reaction needed to generate the C_P_SO_2_H is reversible through an ATP-dependent sulfiredoxin (Srx), whereas the reaction that generates the C_P_SO_3_H is irreversible [11,17]. The overoxidation of this thiol promotes the Prx to restructure and generate decamer-type oligomers (five homodimers, also known as “toroids”). At this point, the antioxidant activity of the Prx diminishes, favoring its transformation into a protein with a chaperone function. Only “typical” Prxs are believed to generate this type of oligomer [44] due to the presence of the motifs sensitive to H_2_O_2_, which are absent in *Ts*Prx1 (Figure 1). This suggests that *Ts*Prx1 could be a robust Prx similarly to that of the Prx (AhpC) of *Salmonella typhimurium* [19].

As previously mentioned, the low peroxidase activity of the Prxs could be related to the fact that the catalytic residues are cysteines [16,17,44], in contrast with those GPx selenium-dependent (GPx-Sec), which are generally more active [47]. The insertion of a Sec residue in a protein through site-directed mutagenesis enables enzymes to use a greater spectrum of substrates, including H_2_O_2_. In addition, the substitution of the essential serine residue by a Sec residue (Ser/Sec) in the subtilisin protease led to a loss of its original activity and the acquisition of a peroxidase activity [48]. A similar result was obtained for the GPx-Sec: the substitution of Sec residue with a Cys residue drastically reduced its activity and increased its sensitivity to overoxidation by H_2_O_2_ [49]. In our study, we found that *Tc*TGR possesses an essential Sec residue that is likely responsible for its high peroxidase activity (Figure 4). Calculations of the initial velocity, using H_2_O_2_ as a substrate, revealed a 20 times higher activity compared to the activity of *Ec*TrxR. These data support the important role of the Sec residue in the peroxidase activity of this enzyme.

The results shown in Figure 3 and Table 3 reveal that *Tc*TGR contributes greatly to reduce H_2_O_2_. This suggests that when the assay system contains TGR and Prx, the peroxidase activity observed at low H_2_O_2_ concentrations is due mainly to *Ts*Prx1 and *Ts*Prx3, whereas at high concentration of the peroxide, where *Ts*Prxs are already saturated, the reducing activity must be attributed to *Tc*TGR. Additionally, in the intact organism, the TGR and its corresponding Prx coexist physiologically and are present in both cytosol and mitochondria; hence, their relative participation in peroxides depuration will depend not only on their kinetic parameters but also on the concentration in each organelle. In this sense, it is well known that the peroxiredoxins represent an important fraction of the total protein in a large variety of organisms, reaching up to 1% of the total soluble protein [44]. In this case, in the cestodes, it will be necessary to assess the concentration of these enzymes in intact organisms to obtain conclusive evidence about their relative importance in H_2_O_2_ depuration. We found no significant differences in the kinetic parameters for *Ts*Prxs1 and *Ts*Prx3, under the conditions used in this study. However, we found differences between the two Prxs in the GS protection assay (Figure 5). *Ts*Prx3 does not protect GS from oxidative damage, possibly because this isoform harbors the motifs sensitive to overoxidation. On the other hand, *Tc*TGR and *Ts*Prx1 do protect GS from oxidative damage.

As mentioned in the Introduction, *T. crassiceps* cysticerci can tolerate high H_2_O_2_ concentrations in the millimolar range [28,29]. Although, under physiological conditions it is barely probable to reach such levels, the kinetic characteristics described for *Tc*TGR (*K*_m_ ~200 μM by H_2_O_2_ and *V*_max_ ~10.36 μmol min^−1^ mg^−1^) seem to have evolved to work in the presence of moderately high H_2_O_2_ concentrations. *Ts*Prxs1 and *Ts*Prx3 have significantly higher affinities for the peroxide, compared with *Tc*TGR (Table 3), which suggests that these enzymes constitute the first in vivo line of defense to avoid oxidative damage. Although using the *Tc*TGR of the parasite in the present work as a coupling enzyme exceeded the activity of *Ts*Prxs, its presence in the enzymatic assays reflects a situation closer to the physiological conditions of the parasite where the three enzymes act in the presence of the others. Hence, in these types of parasites, two very efficient systems have evolved for removing H_2_O_2_, one cytosolic represented by the cytosolic TGR and Prx1 and another mitochondrial that involves the mitochondrial TGR variant and Prx3.

## 5. Conclusions

The high peroxidase activity of TGR within TS could explain two relevant aspects in the physiology of the *T. crassiceps* cysticerci: (a) the tolerance of the parasite to millimolar H_2_O_2_ concentrations [29] and (b) the lack of the CAT gene in trematodes and cestodes [7]. The Prx/Trx/TGR system would compensate for the catalase activity, highlighting TGR’s role in redox homeostasis in these two groups of parasites.

## Figures and Tables

**Figure 1 antioxidants-13-00444-f001:**
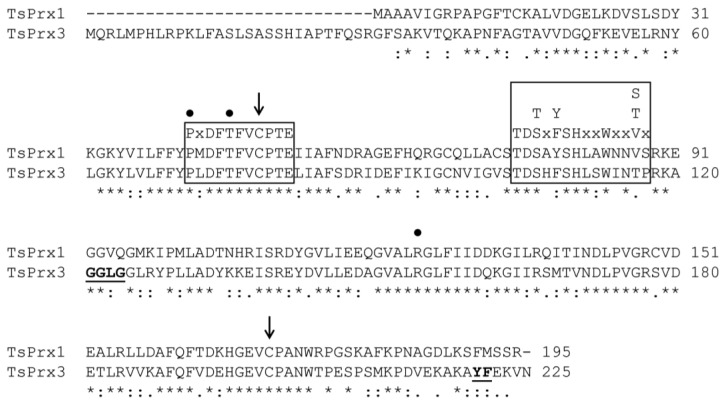
Protein sequence alignment of *Ts*Prx1 and *Ts*Prx3. The alignment was obtained using ClustalO. Symbols indicate (*) identical amino acids, (:) similar amino acids, (.) amino acids with different biochemical properties. Arrows are the cysteine residues (C^49^: peroxidatic cysteine and C^170^: resolving cysteine) involved in the catalytic reaction. In boxes, two highly conserved motifs in the “typical” 2-Cys Prx. Points are residues involved in the stabilization of C^49^. In bold and underlined, *Ts*Prx3-containing residues involved in the hyperoxidation of H_2_O_2_.

**Figure 2 antioxidants-13-00444-f002:**
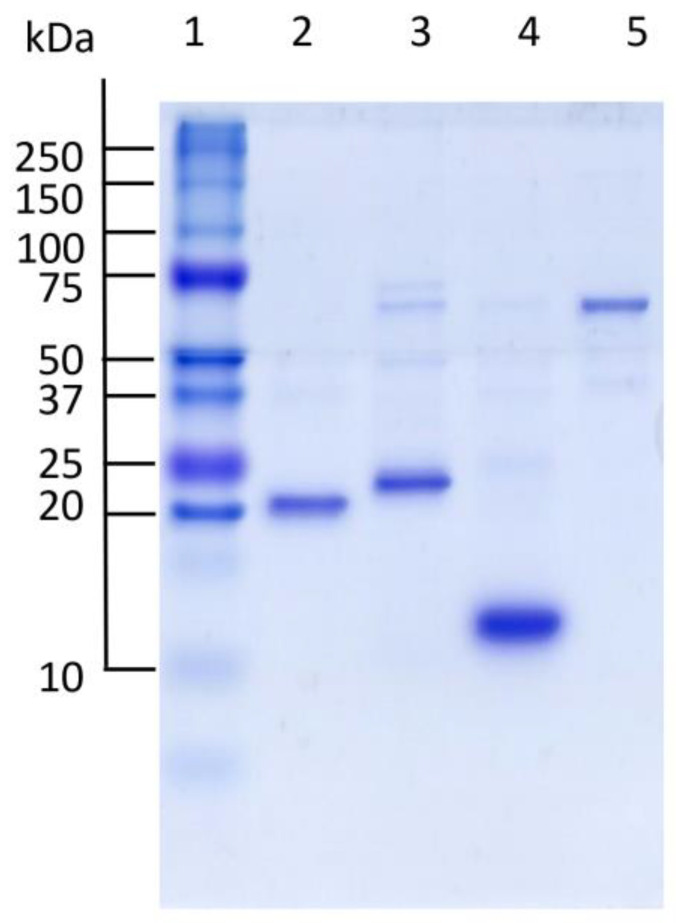
Electrophoretic patterns of thioredoxin system proteins from the *Taenia* genus. Proteins were obtained from the different purification protocols. Lanes are as follows: lane 1, MW markers; lane 2, *Ts*Prx1 (8.0 μg); lane 3, *Ts*Prx3 (10.6 μg); lane 4, *Ts*Trx1 (5.3 μg); lane 5, *Tc*TGR (4.4 μg). Purity grade determined by densitometry for *Ts*Prx1 (87%), *Ts*Prx3 (81%), *Ts*Trx (85%), and *Tc*TGR (75%).

**Figure 3 antioxidants-13-00444-f003:**
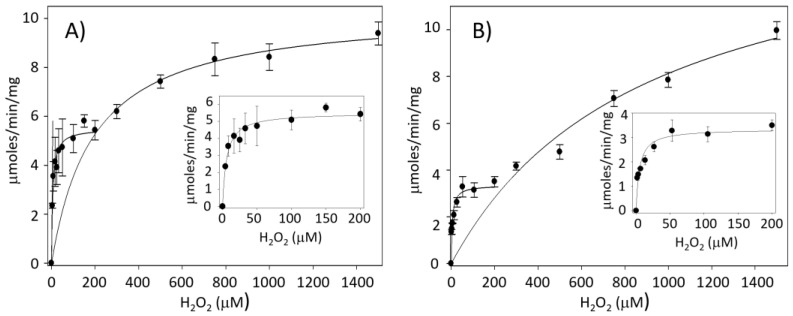
Two enzymes with peroxidative activity. (**A**) *Ts*Prx1 and *Tc*TGR, Michaelis–Menten plot; (insert) magnification of the lower concentrations <200 μM H_2_O_2_ and (**B**) *Ts*Prx3 and *Tc*TGR. The graphs were adjusted to protein concentration of *Tc*TGR [11.2 nM] as well as *Ts*Prx1 and *Ts*Prx3 [1.25 μM]. Data are the means of three independent measurements.

**Figure 4 antioxidants-13-00444-f004:**
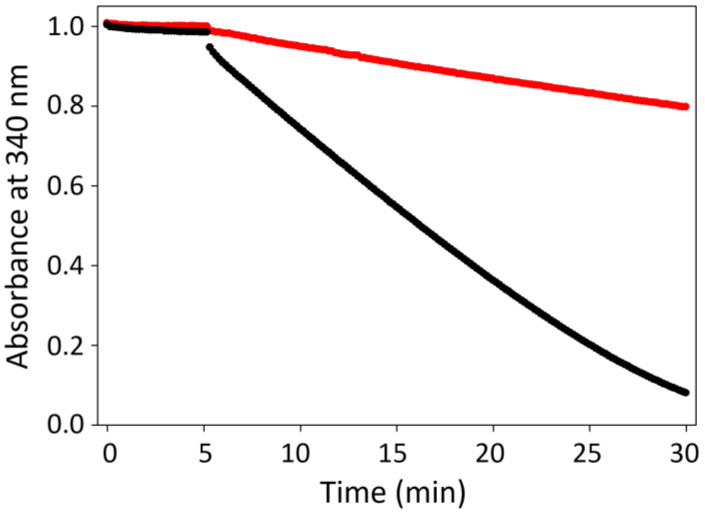
Influence of Sec or Cys residues in hydroperoxide reductase activity. The ability to reduce hydroperoxide of *Tc*TGR-Sec (in black) and *Ec*TrxR-Cys (in red) was determined. Measurements obtained as described under Materials and Methods Section 2. Black line: 11.2 nM *Tc*TGR and 60 μM *Ts*Trx, and red line: 83 nM *Ec*TrxR and 6 μM *Ec*Trx, and 1 mM H_2_O_2_ was added to start the reaction. The decrease in absorbance at 340 nm was recorded.

**Figure 5 antioxidants-13-00444-f005:**
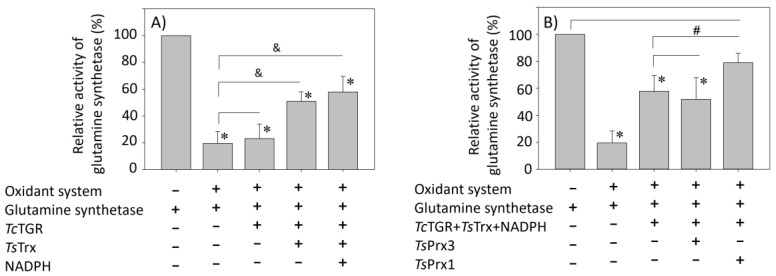
Protection of glutamine synthetase by the different components of thioredoxin system and by the *Ts*Prx1, *Ts*Prx3, and *Tc*TGR enzymes. The different components of TS: *Tc*TGR (11.2 nM), *Ts*Trx (60 μM), NADPH (100 μM), *Ts*Prx1 (1.25 μM), or *Ts*Prx3 (1.25 μM) were incubated with GS from *E. coli* (150 nM) in the presence of a mixed-function oxidation system (OS) in a final volume of 50 μL. After 15 min, 2 mL of the γ-glutamyl transferase assay mixture were added. Additional details are described under Materials and Methods Section 2. (**A**) TS bar 1, positive control; bar 2, negative control; bar 3, mixture without *Ts*Trx and NADPH; bar 4, mixture without NADPH; bar 5, full mixture. (**B**) Enzymes *Ts*Prx1, *Ts*Prx3, and *Tc*TGR. Bar 1, positive control; bar 2, negative control; bar 3, full mixture with *Tc*TGR; bar 4, full mixture with *Ts*Prx3; and bar 5, full mixture with *Ts*Prx1. Statistical significance was considered at a *p*-value < 0.05, as indicated: ∗ = comparison between the different components of the TS vs. GS activity control; & = comparison between the different components of the TS vs. GS residual activity in the presence of the OS; # = comparison between TS vs. *Ts*Prx1 or *Ts*Prx3.

**Table 1 antioxidants-13-00444-t001:** Kinetic constants of recombinant *Ts*Prx1 and *Ts*Prx3 toward H_2_O_2_ in the presence of Trx.

Hydrogen Peroxide				
Enzyme	Reducing Substrate	*K*_m_ (M)	*k*_cat_ (s^−1^)	*k*_cat_/*K*_m_ (M^−1^ s^−1^)
*Ts*Prx1	*Ts*Trx	1.8 ± 0.5 × 10^−6^	160 ± 7.1 × 10^−3^	8.8 × 10^4^
*Ts*Prx3	*Ts*Trx	1.3 ± 0.5 × 10^−6^	90 ± 4.2 × 10^−3^	6.9 × 10^4^

Data obtained using 150 μM NADPH, 11.2 nM TcTGR, 60 μM TsTrx, 1.25 μM TsPrx1, or TsPrx3, and increasing concentrations of H_2_O_2_ at 25 °C and pH 7.8; data are the means of three independent measurements.

**Table 2 antioxidants-13-00444-t002:** Kinetic constants for recombinant *Ts*Prx1 and *Ts*Prx3 for *Ts*Trx.

Enzyme	*Ts*Trx		
	*K*_m_ (M)	*k*_cat_ (s^−1^)	*k*_cat_/*K*_m_ (M^−1^ s^−1^)
*Ts*Prx1	38.6 ± 1.8 × 10^−6^	160 ± 4.0 × 10^−3^	4.1 × 10^3^
*Ts*Prx3	122.0 ± 14.5 × 10^−6^	100 ± 7.3 × 10^−3^	0.8 × 10^3^

Measurements obtained as described under Materials and Methods Section 2; data are the means of three independent measurements.

**Table 3 antioxidants-13-00444-t003:** Kinetic constants for H_2_O_2_ reduction by recombinant *Ts*Prx1 and *Ts*Prx3 and by *Tc*TGR.

Hydrogen Peroxide				
Enzyme	Thioredoxin System	*K*_m_ (M)	*k*_cat_ (s^−1^)	*k*_cat_/*K*_m_ (M^−1^ s^−1^)
*Ts*Prx1 *	*Tc*TGR + *Ts*Trx	5.8 ± 1.0 × 10^−6^	64 ± 2.1 × 10^−3^	1.0 × 10^4^
*Ts*Prx3 **	*Tc*TGR + *Ts*Trx	4.9 ± 1.2 × 10^−6^	35 ± 1.9 × 10^−3^	0.7 × 10^4^
*Tc*TGR *	*Ts*Prx1 + *Ts*Trx	192.0 ± 16.1 × 10^−6^	11,200 ± 230.0 × 10^−3^	5.8 × 10^4^

Data obtained using 150 *μ*M NADPH, 60 μM *Ts*Trx, 11.2 nM *Tc*TGR, 1.25 μM *Ts*Prx1, and increasing concentrations of H_2_O_2_ (2–1500 μM) at 25 °C and pH 7.8; * data obtained from the Michaelis–Menten graph of two enzymes with peroxidase activity (Figure 3A and insert). Lines one and three (cytosolic *Ts*Prx1 and *Tc*TGR); ** data obtained from the Michaelis–Menten graph of two enzymes with peroxidase activity (Insert, Figure 3B). Line two (mitochondrial *Ts*Prx3).

## Data Availability

The authors declare that all data supporting the findings of this study are available within the article and its Appendix A.

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
