# Peer review of "A Physiological Approach to Explore How Thioredoxin–Glutathione Reductase (TGR) and Peroxiredoxin (Prx) Eliminate H2O2 in Cysticerci of Taenia"

_antioxidants, 2024, doi:10.3390/antiox13040444_

Round 1
Reviewer 1 Report
Authors present an interesting if challenging manuscript detailing the peroxiredoxins of several organisms and these enzymes' activities towards hydrogen peroxide. Authors categorize/characterize these different enzymes and conclude that these different enzymes have different roles in the physiology of these organisms, most interestingly that the classical role of catalase may be fulfilled by TGR in T. crassiceps. It appears to be original/novel work on these organisms and authors have supported their hypothesis in specific ways that may be interesting to the readers of Antioxidants. I have the following concerns:
1) Much of the introduction is very very dense and the arc of the manuscript and its story about peroxide removal systems relating to these enzymes and the interesting features of the enzyme can be heard to hear with so much information packed in so tightly. The title suggests a "physiological approach" for exploration of peroxide removal in a specific group of organisms, but I wonder if it might be more specific and if the introduction might focus more intently on the primary findings (i.e. those two major conclusions at the end of the manuscript)? I lack the expertise and context to suggest modulations of the title, but firmer connections between it and the conclusions would help the reader follow the arc of the manuscript.
2) Similarly, the discussion is very detailed in certain ways that are excellent, but drawing broader themes from the work can be challenging with complex discussions on some other minutia that may or may not be valuable. Often for any given discussion paragraph, I initially am able to follow, somewhere in the middle I might get somewhat lost, but then the bottom line (of the paragraph) I am able to agree with and mostly understand/support. Since I'm not usually considering the catalytic activity of Prxs, to some extent less is more and I need to be told more clearly what is important and exciting. A useful example might be the affinity discussion near the end: I think I remember from the good old days that affinity for peroxide is less relevant at high concentrations, thus the high efficiency of TGR despite its lower affinity is interesting/compelling and supportive of the authors' hypotheses. That kind of simple, bottom line, general admissions-level relating of the data to each other to give the reader a take-home insight then gets lost in the subsequent three or four sentences of that paragraph discussing the mitochondrial variants (etc) that feel somewhat extraneous and less pithy.
3) I'm not familiar with the oxidation system utilized but I did read the paper cited. I think readers would benefit from a bit more methodological detail on that oxidation system and then unclear whether it has subsequently studied in greater detail to demonstrate rates of peroxide production in greater detail/precision that might be commented on in the manuscript in useful ways for the reader? I have some sense that this system would certainly produce peroxide, but the reader would have to dig for that. I'm ignorant, maybe this oxidation model is very very common? But (for me) more specific description and context on this feature would be helpful given the technical and chemical detail present in the manuscript.
1) Figure 1, I see when I look closely that there is color there, but its hard to pick out. Unclear how important the homology may or may not be? Is it of value to put the "classic" or "typical" active sites as a third line in those boxes maybe? Consider highlighting or some other way to draw out the residues of interest.
2) I'm unclear how the purification numbers for figure 2 were generated exactly? How was TsPrx1 determined to be 87% pure from that SDS-PAGE?
3) For Figure 5, are there error bars on those measures, statistical descriptions of the results?? This is an important feature of the study that seemed to play less prominently in discussion than I anticipated.
4) Methods state that standard error was used, why is that? I would defer to a statistical reviewer about what's appropriate here, but found this atypical.
Author Response
Please see the attachment included.

Reviewer 2 Report
Please see the attached file.
Please see the attached file.

Author Response
Please see the attachment included

Reviewer 3 Report
In the work entitled “A physiological approach to explore how thioredoxin-glutathione reductase (TGR) and peroxiredoxin (Prx) eliminate H2O2 in cysticerci of Taenia” the authors set out to study the interplay between Prxs and TGR as players in the antioxidant defense in cysticerci of the Taenia genus. To achieve this, two Prxs genes from T. solium were cloned and the respective proteins were expressed and purified, while TGR was purified from T. crassiceps. These purified enzymes were used in several assays in order to characterize their enzymatic activities, giving insights about their catalytic efficiency and affinity for the different substrates, as well as how they possibly work together to attain the redox homeostasis of the parasites regarding H2O2.
The overall work is very written and the main aim of the study was accomplished. The methodology is appropriate and thorough. The results are presented and discussed in a clear and to-the-point manner without overlooking any of the data, however some although some concerns have arisen regarding the manner in which some concepts are presented.
Generally, very good quality of English, with some inconsistencies along the text or a sentence more difficult to understand, here and there.
Major revisions:
1) Lines 109-111 – I am unsure if the conclusions of the work should be presented at this point in the manuscript, as it is usually reserved for the aims. Moreover and, more importantly, the conclusions presented here differ from those presented in the abstract, as in the latter only TsPrx1 is mentioned and here both TsPrx1 and TsPrx3 are cited. The authors must clarify this issue.
2) Lines 212, 329-330, 417, 429, Table 1 and Table S2 – I find slightly dubious the use of the term “substrate” in reference to Trx and GSH in relation to Prxs, as their “true” substrates are H2O2 and hydroperoxides, and the former ones are regenerators of reduced Prxs. I suggest for the authors to change this terminology to better encompass this notion. Perhaps something like “reducing agent”, “redundant” or “reducer”.
3) Lines 316-322 – I suggest changing the order of this paragraph to appear after the next one (lines 329-333), since I believe that it will improve the reading experience of the text.
Other revisions:
1) Lines 12, 20, 21, 54, 68, 70, 98, 99, 142, 197, 204, 216, 402-405, 473, 500, 501 and line 26 of Suppl. Mat. – There are several inconsistencies along the text regarding abbreviations. Some abbreviations are not necessary as the full word only appears once, others are not described with a full word and, many times across the text the full word is used after the abbreviation has been introduced (sometimes the abbreviation description again).
2) Lines 67, 68, 93 – I suggest cutting “the” before Trx, GSH and one hand, respectively.
3) Line 84 – I suggest rewriting the sentence as “… systems being the most…”
4) Lines 87-88 – I suggest rewriting the sentence as “… to recognize as substrate Trxs of another origin.”
5) Line 279 – I suggest cutting “contained”.
6) Line 430 – I suggest ending the sentence as “…GSH more efficiently.”
7) Line 460 – Replace “…of each…” by “…in each…”.
8) Line 462 – I suggest cutting “this is”.
9) Line 27 of Supl. Mat. – I suggest cutting “plus”.
Author Response
Please see the attachment included

Round 2
Reviewer 1 Report
Authors have been very responsive, no further concerns.
None.
Author Response
Thank you for the revision.
Reviewer 2 Report
n/a
Abstract (Lines 22-23): it is still confusing. TcTGR has a bigger Km (192 uM), meaning a weaker, rather than greater, H2O2 affinity compared to that for Prx1 and Prx3.
Author Response
Thank you for the revision.
Lines 22-23, which were confusing, have been modified and we think they are clearer now.
Reviewer 3 Report
The revised version of this manuscript is greatly improved as the authors responded to all queries from the reviewer and performed the necessary changes to the article.
No comments.
Author Response
Thank you for the revision.
The manuscript has been checked and improved as you suggested. Changes are highlighted in yellow in the attached manuscript.